# Research on a Density-Based Clustering Method for Eliminating Inter-Frame Feature Mismatches in Visual SLAM Under Dynamic Scenes

**DOI:** 10.3390/s25030622

**Published:** 2025-01-22

**Authors:** Zhiyong Yang, Kun Zhao, Shengze Yang, Yuhong Xiong, Changjin Zhang, Lielei Deng, Daode Zhang

**Affiliations:** 1Engineering Research and Design Institute of Agricultural Equipment, Hubei University of Technology, Wuhan 430068, China; yzy017@126.com; 2Hubei Engineering Research Center for Intellectualization of Agricultural Equipment, Wuhan 430068, China; 3School of Mechanical Engineering, Hubei University of Technology, Wuhan 430068, China; 102210173@hbut.edu.cn (K.Z.); 102312493@hbut.edu.cn (S.Y.);; 4Hubei Key Laboratory Modern Manufacturing Quality Engineering, School of Mechanical Engineering, Hubei University of Technology, Wuhan 430068, China

**Keywords:** VSLAM, DBSCAN, feature matching, improved RANSAC

## Abstract

Visual SLAM relies on the motion information of static feature points in keyframes for both localization and map construction. Dynamic feature points interfere with inter-frame motion pose estimation, thereby affecting the accuracy of map construction and the overall robustness of the visual SLAM system. To address this issue, this paper proposes a method for eliminating feature mismatches between frames in visual SLAM under dynamic scenes. First, a spatial clustering-based RANSAC method is introduced. This method eliminates mismatches by leveraging the distribution of dynamic and static feature points, clustering the points, and separating dynamic from static clusters, retaining only the static clusters to generate a high-quality dataset. Next, the RANSAC method is introduced to fit the geometric model of feature matches, eliminating local mismatches in the high-quality dataset with fewer iterations. The accuracy of the DSSAC-RANSAC method in eliminating feature mismatches between frames is then tested on both indoor and outdoor dynamic datasets, and the robustness of the proposed algorithm is further verified on self-collected outdoor datasets. Experimental results demonstrate that the proposed algorithm reduces the average reprojection error by 58.5% and 49.2%, respectively, when compared to traditional RANSAC and GMS-RANSAC methods. The reprojection error variance is reduced by 65.2% and 63.0%, while the processing time is reduced by 69.4% and 31.5%, respectively. Finally, the proposed algorithm is integrated into the initialization thread of ORB-SLAM2 and the tracking thread of ORB-SLAM3 to validate its effectiveness in eliminating feature mismatches between frames in visual SLAM.

## 1. Introduction

Visual SLAM enables self-localization and environmental mapping by capturing and processing feature information from the environment [1]. Dynamic features from moving objects may be incorrectly identified as part of the environment, resulting in localization errors and incorrect map information in visual SLAM. The ability to accurately eliminate dynamic features during the inter-frame matching process is critical to the robot’s localization accuracy and the quality of the environmental map construction. Camera hardware is cost-effective, adaptable, and provides rich image data, making it widely utilized in visual SLAM [2]. A major challenge in visual SLAM is the efficient and accurate removal of dynamic features while processing environmental data in dynamic environments.

Feature matching errors are among the primary causes of performance degradation in visual SLAM [3]. The Random Sample Consensus (RANSAC) method is widely used in visual SLAM to effectively eliminate inter-frame mismatches. However, when a dataset contains numerous outliers, the RANSAC method necessitates more iterations to compute an accurate geometric model. To address this issue, researchers have proposed several improvements. Raguram et al. [4] introduced the USAC method, which improves inlier accuracy and efficiency through modular design and various optimization techniques. The MLESAC method proposed by Torr et al. [5] enhances robustness by optimizing the probability function for model fitting. Chum et al. introduced the PROSAC [6] and LO-RANSAC [7] methods. PROSAC prioritizes sampling based on match-points confidence to improve algorithm speed, while LO-RANSAC incorporates local optimization steps to enhance inlier quality. He et al. [8] proposed an adaptive error threshold RANSAC method determined by inlier matching rates, which improves the algorithm’s accuracy in eliminating feature mismatches in both dynamic and static scenes. Although these methods have improved algorithm performance to some extent, they primarily rely on local neighborhood information of feature points in images, lacking global constraints. This makes them less efficient in high-noise data, dynamic scenes, or under complex geometric constraints [9], and they still face challenges in effectively eliminating mismatched points in dynamic environments. For image pairs with structurally similar or repetitively textured scenes in dynamic environments, the initial feature matching set often contains a large number of mismatches, significantly impacting the accuracy of subsequent camera pose estimation and 3D reconstruction in visual SLAM systems.

To address these issues, the main contributions of this study are as follows:This paper proposes a density-based RANSAC method (DSSAC), which pre-processes the dataset through clustering and dividing it into dynamic and static clusters to accurately eliminate dynamic feature points. Since the data are pre-clustered, the required number of iterations is significantly reduced compared to traditional RANSAC. This method combines geometric analysis and clustering techniques, offering an efficient and accurate strategy for elimination mismatches.RANSAC, as a highly robust fitting method, excels in fitting high-quality data. By combining the DSSAC method with RANSAC, the former effectively removes noise and clustered dynamic points through clustering, while the latter is employed for secondary screening of mismatched points, achieving higher accuracy and computational efficiency.This study evaluates the performance of the DSSAC-RANSAC method and verifies its effectiveness in eliminating mismatches in indoor and outdoor dynamic scenes. Using reprojection error mean, reprojection error variance, and processing time as evaluation metrics, experimental results demonstrate that DSSAC-RANSAC significantly outperforms traditional methods in eliminating dynamic feature points. The algorithm is applied to the initialization thread of ORB-SLAM2 and the tracking thread of ORB-SLAM3 to validate its feasibility within visual SLAM systems.

Section 2 reviews related studies on feature mismatch elimination methods, analyzing their applications and limitations in feature matching and mismatch removal. Section 3 provides a detailed explanation of the proposed improved algorithm, including the core concept, algorithm design, and implementation details. Section 4 presents the experimental section, where the algorithm’s performance is tested on public datasets, and the applicability and robustness of the method are verified using self-collected datasets. The results are systematically organized and analyzed for clarity and insight. Additionally, the proposed algorithm is integrated into the initialization thread of ORB-SLAM2 and the tracking thread of ORB-SLAM3 to further evaluate its practical application. Section 5 concludes the paper and discusses future research directions, as well as potential improvements.

## 2. Related Works

Random Sample Consensus (RANSAC) is a widely used method for data fitting. It seeks a mathematical model that best fits the data characteristics by randomly selecting points from the sample and evaluates the fitting quality using reprojection error, iterating to identify the optimal model [10]. Fischler M.A. et al. [11] used RANSAC to eliminate mismatched feature points in image feature matching by randomly selecting four pairs of matching points from images A and B, then solving the homography matrix *H* using the Direct Linear Transform (DLT) method:(1)$$c\left(\begin{array}{l} u\\ v\\ 1 \end{array}\right)=H\left(\begin{array}{l} u^{\prime }\\ v^{\prime }\\ 1 \end{array}\right)$$

In Equation (1), where *c* is a constant,$p={\left(u\;\;\;v\;\;\;1\right)}^{t}$ and p is a point in image A, $p^{\prime }={\left(u^{\prime }\;\;\;v^{\prime }\;\;\;1\right)}^{t}$ and $p^{\prime }$ is a point in image B, p and $p^{\prime }$ form a matching pair, and $H=\left(\begin{array}{l} h_{1}\;\;\;h_{2}\;\;\;h_{3}\\ h_{4}\;\;\;h_{5}\;\;\;h_{6}\\ h_{7}\;\;\;h_{8}\;\;\;h_{9} \end{array}\right)$. After solving for the homography matrix, the reprojection error between the matching feature points of the two images is calculated and compared to a predefined reprojection error threshold. The suitability of the obtained homography matrix is then determined based on the inlier ratio. Subsequently, points are iteratively selected until the optimal homography matrix *H* and the inlier set are obtained within a predefined number of iterations.

Density-Based Spatial Clustering of Applications with Noise (DBSCAN) is a density-based spatial clustering method [12]. Unlike other clustering methods, DBSCAN eliminates the need to predetermine the number of clusters and can detect clusters of arbitrary shapes, leading to its extensive application in various domains [13]. In an improvement regarding adaptive parameter estimation, Jiang et al. [14] observed that correct matches during feature matching typically exhibit similar motion patterns, which can be grouped into several motion consistency groups, whereas incorrect matches tend to be randomly distributed in the image domain. Therefore, they modeled the feature matching process in visual SLAM as a spatial clustering problem.

During the clustering process, the neighborhood threshold r and the minimum number of points *MinPts* are critical parameters affecting clustering quality.

First, the weighted distance is used to compute the distance *D* between each pair of feature points. If feature points $p_{i}$ and $q_{i}$ exist in image A, and their corresponding feature points in image B are $p_{j}$ and $q_{j}$, the distance *d* is calculated as:(2)$$d_{pq}=d(p_{i},q_{i})+d(p_{j},q_{j})+\omega_{ij}\times d(p_{ij},q_{ij})$$

In Equation (2), the weight coefficient *wij* is defined as:(3)$$w_{pq}=1+\gamma \times e^{-min\{d(p_{i},q_{i})+d(p_{j},q_{j})\}}$$

In Equation (3), where $d_{pq}$ represents the weighted distance between two pairs of feature points, $d(p_{i},q_{i})$ denotes the Euclidean distance between feature points $p_{i}$ and $q_{i}$, and γ is a positive parameter enhancing motion consistency.

If there are *N* pairs of matching points, an *N × N* distance matrix *D* can be constructed after calculating all $d_{pq}$ values. The neighborhood distance *r* is calculated based on the maximum value *MA* and minimum value *MI* of *D* using the following formula:(4)r=λ(MA−MI)+MI

In Equation (4), λ is a scaling parameter used to adjust the size of the neighborhood radius. The minimum number of points within a neighborhood, minpts, is defined as:(5)MinPts=N×pct

In Equation (5), *N* represents the total number of matching point pairs, and *pct* is the predefined percentage, typically ranging from 3% to 5%.

In the point set $U=\{p_{1},p_{2},\ldots ,p_{n}\}$, for any point p∈U, if p satisfies the following conditions:(6)$$\left\{\begin{array}{l} N_{r}(p)=\left\{q\in U|dist(p,q)\leq r\right\}\\ \left|N_{r}(p)\right|\geq MinPts \end{array}\right.$$

The point p is defined as a core point if it meets the criteria. In Equation (6), dist(p,q) represents the distance between point *p* and point *q*, *r* is the defined neighborhood distance, and *MinPts* is the minimum number of points required within the neighborhood to qualify *p* as a core point.

If the number of points in the neighborhood of point *p* is less than *MinPts*, but it lies within the neighborhood of a core point, *p* is defined as a border point. Border points do not participate in the expansion of new clusters but are assigned to the cluster of their neighboring core points. Points that neither belong to the neighborhood of any core point nor meet the minimum point requirement are defined as noise points and are removed after clustering.

The clustering process of DBSCAN is performed by traversing the dataset. Starting with the first point that meets the core point criteria, its cluster is expanded by checking whether its neighboring points also satisfy the core point condition. If so, these points are added to the current cluster, and the process continues until no new core points can be identified. After traversing all points, those that do not belong to any cluster and do not meet the core point condition are labeled as noise points.

Liu et al. [15] adaptively calculated parameters using a k-distance graph before clustering, improving the robustness of the clustering method in handling datasets with varying densities. Bryant et al. [12] integrated RNN methods to precompute adaptive parameters based on the distribution of the data, enhancing the robustness of the method. Liu et al. [16] used KD-trees to optimize the traversal process for cluster expansion, thereby improving the execution speed of the clustering algorithm. Hu et al. [17] leveraged global information from feature matching to construct a feature matching set, using clustering methods to eliminate noise points within matching clusters, effectively removing mismatched pairs.

Although the aforementioned improvements significantly enhance the mismatch elimination capability of the DBSCAN method in static environments, certain limitations persist in dynamic environments. Specifically, multiple adjacent matching point pairs may appear on the surface of dynamic objects. Due to spatial proximity, these point pairs might be incorrectly classified into the same cluster, thus interfering with the correct distinction between dynamic and static features. Such misclassification significantly affects the subsequent pose estimation and map construction accuracy in the visual SLAM system, limiting its robustness and adaptability in dynamic scenarios. Inspired by this, this study combines DBSCAN and RANSAC methods to propose a density-based RANSAC method suited for dynamic environments. In dynamic scenarios with a large number of outliers, this method can accurately eliminate outliers, identify and remove dynamic clusters, and efficiently obtain a set of static matching pairs.

## 3. Methodology

### 3.1. DSSAC-RANSAC Framework

This paper proposes a density-based RANSAC method, which applies RANSAC three times to eliminate mismatches. The method introduces global clustering, enabling more accurate image feature matching in dynamic scenes by processing the initial set with density clustering, thereby significantly improving the algorithm’s speed. Since ORB is simpler and more efficient than SIFT, SURF, and other feature detectors [18], with advantages such as rotational invariance, it is widely used in real-time image processing and computer vision. In the following discussion, ORB is used to illustrate the algorithm.

The flowchart of DSSAC-RANSAC is shown in Figure 1, and the algorithm flow is as follows:

1.Capture and output the corresponding 2D images $P_{A}$ and $P_{B}$ through the camera.2.Detect key points in images $P_{A}$ and $P_{B}$, calculate the descriptors of the key points, and output the ORB feature points.3.Perform brute-force matching of ORB features based on Hamming distance [19], obtaining the initial matching set $C_{BF}$.4.Perform down-sampling on the sample set to obtain the matching set $C_{D}$.5.Apply the DSSAC method proposed in this paper for local clustering, obtaining the clustered set $C_{DJ}$, and then remove dynamic features to obtain the static point set $C_{M}$.6.Use the RANSAC method to perform local optimization on the point set $C_{M}$.

### 3.2. Density-Based Segmentation RANSAC

Traditional RANSAC is a method that randomly selects points from a dataset to establish a mathematical model, obtaining the best fit by minimizing the reprojection error. The quality of the selected points significantly impacts the fitting performance of the mathematical model. Due to the randomness of point selection, traditional RANSAC requires a large number of iterations to achieve satisfactory matching results, particularly in scenarios with a high number of dynamic features. For different transformations in various scenarios, correct feature matches tend to exhibit coherent motion, with neighboring points sharing similar motion [14]. Based on this characteristic, this paper proposes a density-based segmentation RANSAC (DSSAC) method to preprocess the sample set, establish a reprojection error threshold and inlier ratio, and eliminate dynamic feature points.

DBSCAN is used to perform clustering on all motion vectors, generating multiple clusters. Dense clusters typically contain feature matches with similar motion vectors, while sparse clusters or outliers correspond to mismatches. The algorithm framework is illustrated in Figure 2. Similar to the DBSCAN method, it constructs a feature matching set as follows:$$D={\left\{x_{i}^{T},y_{i}^{T},m_{i}^{T}\right\}}^{T},(i=1,2,\cdots ,N)$$

Here, *D* represents the dataset containing the matched point coordinates and motion vectors of two images. $x_{i}$, $y_{i}$ are 2D vectors representing the spatial positions of the matching points $p_{i,j}$ and $q_{i,j}$ in images $P_{A}$ and $P_{B}$, and $m_{i}$ represents the motion vector of the two matched feature points.

The sample set is divided into distinct clusters based on spatial density, and the resulting feature point clusters are defined as:$$C=\{C_{-1},C_{1},C_{2},\ldots ,C_{K}\}$$

Each cluster $C_{i}$ contains several feature point pairs with similar motion vectors, and i=−1 represents the mismatch cluster.

In the cluster $C_{i}$, the number of matched feature point pairs N>4, which allows at least 4 pairs of feature points $\left(p_{i,j},q_{i,j}\right)$ to be randomly selected as samples for estimating the homography matrix H of the cluster. Using these sample point pairs, the optimal homography matrix H is obtained by minimizing the geometric error between the matching pairs. Project the position $p_{i,j}$ of each feature point pair in C from image $P_{A}$ to image $P_{B}$ to obtain the projected position $q_{i,j}$, then calculate the reprojection error $e_{i,j}$ between $q_{i,j}$ and the actual matched point, as shown in Equation (7):(7)$$e_{i,j}=∥{\hat{q}}_{i,j}-q_{i,j}∥$$

Set a reprojection threshold τ. If $e_{i,j}<\tau$, the point is marked as an inlier. After calculating the reprojection errors for all matched points, the number of inliers n that meet the conditions is counted, and the inlier ratio of the cluster is calculated as follows:(8)$$\alpha =\frac{n}{N}$$

In Equation (8), n is the number of inliers that meet the conditions, N is the total number of matches, and α is the inlier ratio of the cluster. To determine the dynamics of a cluster, an inlier ratio threshold $\alpha_{min}$ is set. If the inlier ratio of cluster $C_{i}$, $\alpha >\alpha_{min}$, the cluster is classified as a static cluster; otherwise, all feature point pairs within the cluster are marked as noise and excluded from the static feature set. This step effectively eliminates the feature point pairs of dynamic clusters while retaining the feature points in static clusters, thereby reducing the impact of dynamic scenes on the localization accuracy of the visual SLAM system. The specific algorithm flow is outlined as follows:

1.Compute the distance matrix and cluster the point set into $C=\{C_{-1},C_{1},C_{2},\ldots ,C_{K}\}$.2.Initialize parameters: set the reprojection error threshold τ, the minimum inlier ratio $\alpha_{min}$, the matching set D, and initialize the number of inliers n=0.3.Select four matching pairs from $C_{i}$ and compute the homography matrix $H_{i}$.4.Using the homography matrix $H_{i}$, calculate the number of inliers n in which the reprojection error $e_{i,j}$, between image $P_{A}$ projected to image $P_{B}$ is less than τ.5.Calculate the inlier ratio α of the cluster by using the total number of matches N and the number of inliers n that meet the conditions.6.If $\alpha >\alpha_{min}$, classify the cluster as a static cluster.7.Iterate through all clusters $C_{i}$ in the dataset C based on their labels, repeat steps (3) to (6), and add all static points to the set M.

This algorithm preprocesses the dataset by clustering the points and computes the homography matrix to eliminate dynamic clusters. It considers both the initial positions and motion vectors of feature points to cluster matches exhibiting consistent motion. Since the data have been preprocessed, satisfactory results can be achieved with fewer iterations when calculating the reprojection error. Since there is another selection step later, the reprojection error threshold τ should be set relatively high. If τ is set too low, it may result in insufficient inliers. Generally, τ is chosen within the range of 50–100. To account for specific environments, such as images containing many dynamic features, the minimum inlier ratio $\alpha_{min}$ should be set lower. If set too high, static clusters may be misclassified as dynamic, thereby affecting the accuracy of the method. Experimental results suggest that setting $\alpha_{min}$ to a value between 0.3 and 0.5 produces better results. It is worth noting that the traditional RANSAC method requires at least four inliers to compute the transformation matrix, while the clusters obtained through clustering already satisfy the minimum inlier requirement. Therefore, there is no need to evaluate the inlier count when selecting points from the clusters. The algorithm flow of DSSAC is outlined as follows:

### 3.3. Image Feature Mismatch Elimination Method Combining DSSAC and RANSAC

After the DSSAC method eliminates dynamic features and significant mismatches, a small number of mismatches may remain in the dataset, potentially affecting the quality of matched points. Although the RANSAC method is robust in the feature matching and effectively handles outliers, it relies on predefined mathematical models, which can reduce its efficiency and accuracy in environments with numerous dynamic features or a large number of outliers. The dataset processed by the DSSAC method eliminates most dynamic features and outliers, thereby significantly reducing interference during the subsequent model fitting process. This enables RANSAC to rapidly and accurately estimate geometric models on high-quality datasets. By combining DSSAC with RANSAC, the DSSAC method effectively removes dynamic features and significant mismatches through clustering, providing more accurate matched point data for subsequent processing. RANSAC further eliminates residual mismatches with fewer iterations, thereby ensuring the robustness and accuracy of geometric model fitting. The combination of these two methods significantly enhances computational efficiency and matching accuracy in dynamic environments. The flow chart of the algorithm in this paper is shown in Figure 3.

In Figure 4, to enhance the observation of details, yellow rectangles are used to highlight regions with variations in feature points, and certain areas are further magnified for clarity. First, perform ORB feature extraction on the input images, followed by brute-force matching of the extracted features (as illustrated in Figure 4a). Second, apply the proposed DSSAC method for global clustering of feature point pairs (with different colors representing different clusters) and initial elimination of mismatches (as shown in Figure 4b), followed by the removal of dynamic feature points (as shown in Figure 4c). Finally, the RANSAC method is employed for local refinement, yielding precise matches (as depicted in Figure 4d).

As illustrated in Figure 4a, brute-force matching methods typically produce a large number of mismatches and dynamic feature points. If RANSAC is directly applied through random sampling iterations, the high probability of selecting mismatched points means that some mismatches may repeatedly appear during multiple iterations, requiring numerous iterations to find the optimal inlier set. Furthermore, each iteration necessitates the calculating of the reprojection error for all matched points, which significantly impacts the algorithm’s efficiency. Therefore, the algorithm proposed in this paper preprocesses the sample set using clustering methods, treating outliers as noise and removing most mismatched points. It classifies feature points based on spatial consistency, effectively identifying and excluding dynamic feature points. However, DSSAC faces challenges in fully removing matched points with small motion vectors that are classified into static clusters. Combining the two algorithms can more effectively eliminate outliers. In the DSSAC algorithm, to ensure the inclusion of all potential static feature point clusters, the reprojection error threshold is set within a relatively large range of 50–100. This relatively loose threshold setting allows for more comprehensive coverage of static features, thereby enhancing the robustness of the data preprocessing stage. The DSSAC algorithm significantly optimizes the initial dataset by effectively eliminating dynamic features. The RANSAC reprojection error threshold is set to 25–50, with the number of iterations limited to 10–20. Since DSSAC preprocessing has significantly reduced the proportion of residual outliers, this configuration with fewer iterations achieves a good balance between accuracy and efficiency.

## 4. Experiments

### 4.1. Datasets and Evaluation Metrics

This study utilizes the TUM [20] and KITTI [21] datasets to conduct experiments on inter-frame image feature matching, with a focus on the DSSAC method in both indoor and outdoor dynamic scenarios to analyze and compare its accuracy and runtime performance. The accuracy tests employ two metrics: the mean error $\left(e_{mean}\right)$ and the error variance $\left(e_{Var}\right)$, defined as:(9)$$e_{mean}=\frac{\sum_{i=1}^{n}\left\|{\hat{q}}_{i,j}-q_{i,j}\right\|}{N}$$(10)$$e_{Var}=\frac{\sum_{i=1}^{n}{\left(e_{mean}-\left\|{\hat{q}}_{i,j}-q_{i,j}\right\|\right)}^{2}}{N}$$

In Equation (9), the mean error $e_{mean}$ represents the average magnitude of all feature point errors and is used to measure the overall fitting accuracy. In Equation (10), the error variance $e_{Var}$ is used to characterize the dispersion of the data, reflecting the stability of the fitting results. These two metrics collectively represent the matching accuracy and robustness of the method in different environments.(11)$$Imprv(E)=\left(1-\frac{E_{our}}{E_{other}}\right)\times 100\%$$

In Equation (11), E represents the evaluation metric, which is used to calculate the percentage improvement of the proposed algorithm relative to other algorithms, thereby quantifying the performance enhancement.

### 4.2. Mismatch Elimination in Different Scenarios

As shown in Figure 5, Figure 6 and Figure 7, two experimental environments were established: an indoor dynamic environment and an outdoor dynamic environment. Yellow rectangles are used to delineate dynamic objects, with specific regions magnified to enhance the comparison of matching results produced by different algorithms. Figure 5 depicts an indoor dynamic environment where two individuals exhibit gesture variations and overall movement. Figure 6 illustrates vehicles moving on an outdoor road. Figure 7 presents self-collected images depicting a queue of individuals on a playground.

ORB feature points were extracted for each experimental scenario. The RANSAC threshold was set to 3, with 50 iterations performed. In GMS-RANSAC, the GMS grid size parameter was set to the default value of 20, with the RANSAC threshold unchanged, and the iteration count reduced to 30 to improve algorithm speed. In GMSATRANSAC, the GMS parameters were kept unchanged, and the internal matching rate was set to 40%. The self-collected dataset shown in Figure 7 was captured using the Intel RealSense D435i camera, with a resolution of 1920 × 1080 and a frame rate of 30 FPS. To enhance image quality, both brightness and contrast were increased by 5%. Monocular mode was employed for data collection to meet the algorithm’s requirements.

As shown in Figure 5, in the indoor dynamic environment, individuals exhibit relatively small movements, and the complex background causes dynamic feature points to be easily confused with the static scene. The RANSAC, GMS-RANSAC and GMS-ATRANSAC algorithms still exhibit feature mismatches on dynamic individuals, where as the DSSAC-RANSAC algorithm more effectively eliminates mismatches in these dynamic features. The DSSAC-RANSAC algorithm uses a clustering method that combines density and distance, leveraging the spatial similarity of dynamic features in indoor scenes to more accurately distinguish dynamic objects from the static background, thereby more effectively removing dynamic feature points. As shown in Figure 6, in outdoor dynamic scenarios, the first two algorithms still produce mismatches on dynamic objects. However, DSSAC-RANSAC, with its dynamic feature recognition mechanism, effectively distinguishes and identifies moving objects, removing their associated feature points during data processing. Therefore, DSSAC-RANSAC more effectively eliminates mismatched feature points in dynamic environments. In the self-collected dataset shown in Figure 7, due to the similarity between dynamic features and the background, the RANSAC and GMS-RANSAC algorithms fail to fully eliminate mismatches related to dynamic individuals. In contrast, DSSAC-RANSAC effectively eliminates all mismatches related to dynamic individuals.

Experimental results show that the DSSAC-RANSAC algorithm exhibits superior robustness in dynamic scenarios, effectively eliminating image feature mismatches in both indoor and outdoor dynamic environments, as well as in the self-collected dataset.

Figure 8 illustrates the mean error and error variance of four algorithms, RANSAC, GMS-RANSAC, GMS-ATRANSAC, and DSSAC-RANSAC, across different numbers of feature points. It is clear that DSSAC-RANSAC consistently maintains low mean error and variance, regardless of the feature point quantity. This demonstrates that DSSAC-RANSAC not only effectively removes outliers but also maintains high fitting accuracy, bringing the matching results closer to the true values.

Figure 9 shows the runtime of four algorithms with different numbers of feature points. RANSAC relies on extensive random sampling to find the optimal inlier set, requiring repeated computations of the model fitting performance. In scenarios with many outliers, the number of samples increases significantly, and each iteration must traverse all feature points, resulting in longer computation times as the number of feature points grows. The GMS-RANSAC and GMS-ATRANSAC methods, although introducing multi-scale matching strategies with multiple scale divisions and iterative evaluations for feature points, require intensive neighborhood matching computations for each scale division and iteration. All three algorithms mentioned above lack a mechanism for removing dynamic features. Feature points on dynamic objects are repeatedly computed, increasing the time cost with each calculation. The DSSAC-RANSAC method uses a dynamic cluster elimination mechanism to effectively identify and remove mismatched clusters on dynamic objects, avoiding redundant computations on dynamic objects. Additionally, uniform down-sampling of samples is performed before clustering, significantly reducing computation time during the matching process and demonstrating a notable speed advantage.

Table 1, Table 2 and Table 3 show the performance of three algorithms under different ORB feature points, including the mean error, error variance, and computation time in both classic datasets and self-collected datasets.

Table 4, Table 5 and Table 6 respectively show the percentage improvement in error methods, mean error difference, time, and other metrics for the DSSAC-RANSAC algorithm with 1500, 2000, and 2500 ORB feature points compared to three other algorithms. From Table 4, Table 5 and Table 6, it can be seen that the DSR algorithm shows significant advantages in all three evaluation metrics for different numbers of feature points.

Table 7 shows the average percentage improvement in error variance, mean error difference, and runtime, among other metrics, for DSSAC-RANSAC compared to the other three algorithms at different feature point quantities. Compared to RANSAC, DSSAC-RANSAC reduced the mean error difference by 58.5% on average and the error variance by 65.2%. Compared to GMS-RANSAC, the mean error difference decreased by 49.2% on average, and the error variance decreased by 63.0%. Compared to GMS-ATRANSAC, the mean error difference decreased by 36.2% on average, and the error variance decreased by 50.6%. In terms of runtime, DSSAC-RANSAC reduced the time by 69.4% on average compared to RANSAC; by 31.5% compared to GMS-RANSAC; and by 13.7% compared to GMS-ATRANSAC. Therefore, DSSAC-RANSAC is a fast and accurate method for eliminating image feature mismatches.

### 4.3. Improvements to the ORB-SLAM3 Feature Matching Module Using the DSSAC-RANSAC Algorithm

The DSSAC-RANSAC algorithm was applied to ORB-SLAM3 to eliminate inter-frame mismatches, improving the monocular initialization thread of ORB-SLAM2 and the tracking thread of ORB-SLAM3. Tests were conducted using the rgbd_dataset_freiburg3_walking_xyz sequence from the TUM dataset.

#### 4.3.1. Monocular Initialization of ORB-SLAM2

Monocular initialization is a critical module in SLAM visual systems, primarily responsible for completing the initial pose estimation. Accurate monocular initialization is crucial for improving the precision and quality of map construction. Since ORB-SLAM3 introduces IMU support, monocular initialization utilizes IMU data for assistance. To focus on the improvement effects of pure vision algorithms, the experiments adopted the monocular initialization framework of ORB-SLAM2. This study applies the DSSAC-RANSAC algorithm to the initialization thread of ORB-SLAM2 and compares it with the default initialization algorithm to evaluate the improvements in inter-frame feature matching.

In the experimental image sequence, the characters wearing a plaid shirt and a black top are both moving in the current frame.In Figure 10, the two ends of the green line segments in the image represent the successfully matched ORB feature points between the previous and current frames, reflecting the motion relationship of feature points between frames. Figure 10a shows the default initialization algorithm of ORB-SLAM2. Due to the movement of the characters and the presence of many similar features on the surface of the plaid shirt, a large number of mismatched points appear on the back. Additionally, the dynamic feature points on the character in a black top on the right were not effectively removed. Keeping these mismatched points during initialization not only significantly affects the accuracy of the initial pose estimation but also negatively impacts the subsequent map building process. Figure 10b shows the improved initialization method proposed in this paper, where the dynamic feature matches on both moving characters were successfully removed, and no mismatched points appeared.

#### 4.3.2. Tracking Thread of ORB-SLAM3

The proposed algorithm was applied to the tracking thread of ORB-SLAM3 to verify its capability to eliminate mismatches in the SLAM system.

In Figure 11a,b, the individual on the left exhibits upper-body and leg movements, while the individual on the right rotates their upper body, the green small rectangles denote the tracked feature points. During the feature matching stage, the DSSAC-RANSAC algorithm effectively distinguishes dynamic features from static background elements, preventing dynamic objects from negatively affecting the camera’s initial pose estimation and map construction. In Figure 11c,d, the red points denote 3D map points derived from feature points preserved through matching across multiple image frames. Figure 11c shows the point cloud map generated by ORB-SLAM3’s default method. It can be observed that the number of near-point clouds is higher and more chaotic due to the influence of dynamic people. Figure 11d presents the depth point cloud map with effective ORB feature points retained, where the features of dynamic individuals near the seats on both sides of the near points are effectively removed. Experimental results show that the algorithm proposed in this paper can robustly and accurately remove dynamic feature points in ORB-SLAM3’s tracking thread, significantly improving the accuracy of local map construction and loop closure detection.

## 5. Conclusions

This paper proposes a density-based improved RANSAC method (DSSAC), which is combined with the traditional RANSAC method. The DSSAC method demonstrates robust outlier elimination capabilities in environments with a high proportion of outliers, effectively distinguishing between dynamic and static clusters to generate high-quality datasets. Building on this, the RANSAC method is employed to further filter a small number of residual outliers. Experimental results show that the DSSAC-RANSAC method significantly reduces mismatches and dynamic feature point interference, achieving higher accuracy and computational efficiency compared to traditional methods. Experiments on different dynamic scenarios demonstrate that the proposed method exhibits superior robustness and accuracy and has been validated on self-collected datasets. Additionally, the DSSAC-RANSAC method was applied to the initialization thread of ORB-SLAM2 and the tracking thread of ORB-SLAM3 for experiments. The results indicate that the method effectively removes dynamic feature points in keyframes, improves the quality of keyframe feature points, and enhances the accuracy and reliability of robot pose estimation and environmental map construction. Future research will focus on further optimizing clustering methods for dynamic features to minimize their impact on static environment features. Additionally, after removing dynamic features, the generated static feature clusters will undergo in-depth analysis, treating each cluster as an independent visual word and integrating its information into the Bag of Words (BoW) model. This improvement is expected to further enhance the processing capability, robustness, and accuracy of visual SLAM systems in dynamic environments.

## Figures and Tables

**Figure 1 sensors-25-00622-f001:**
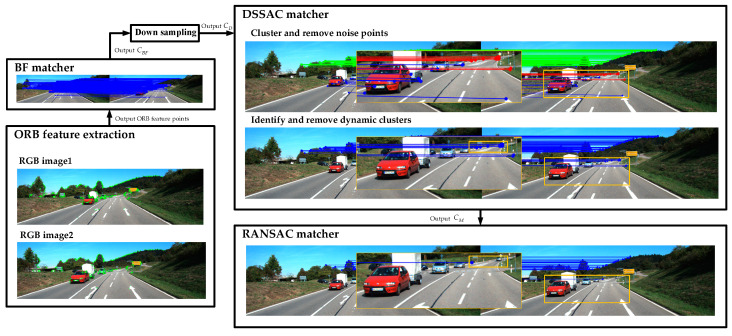
DSSAC-RANSAC algorithm flowchart; the images are derived from the KITTI dataset.

**Figure 2 sensors-25-00622-f002:**
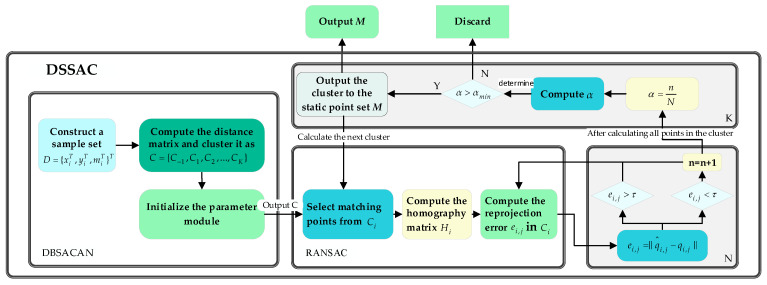
Flowchart of the DSSAC method.

**Figure 3 sensors-25-00622-f003:**
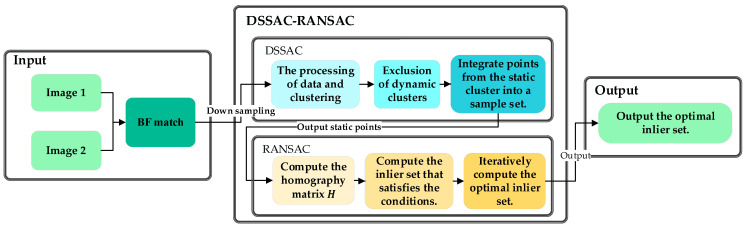
Methodology flowchart of this study.

**Figure 4 sensors-25-00622-f004:**
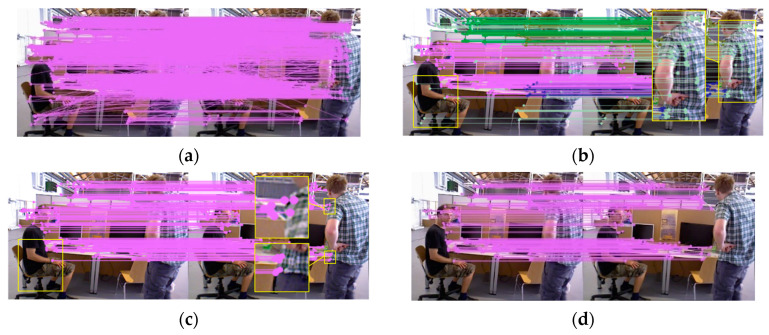
ORB feature matching in the TUM dataset using DSSAC-RANSAC with 1500 feature points. (**a**) Brute-force matching, (**b**,**c**) clustering and dynamic cluster elimination process of DSSAC, and (**d**) results after RANSAC removes a small number of mismatches and low-quality points.

**Figure 5 sensors-25-00622-f005:**
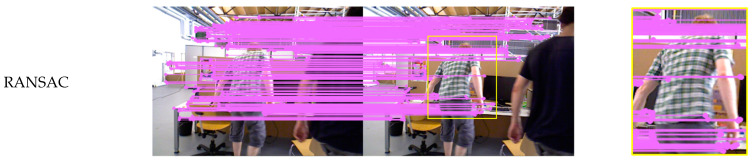
Comparison of feature matching performance between RANSAC, GMS-RANSAC (G-R), GMS-ATRANSAC (G-ATR) and DSSAC-RANSAC (D-R) in TUM indoor dynamic scenes.

**Figure 6 sensors-25-00622-f006:**
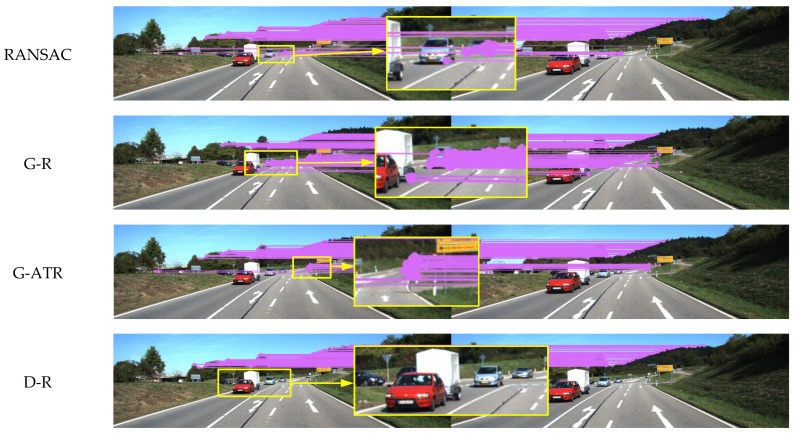
Comparison of feature matching performance between RANSAC, GMS-RANSAC (G-R), GMS-ATRANSAC (G-ATR) and DSSAC-RANSAC (D-R) in KITTI outdoor dynamic scenes.

**Figure 7 sensors-25-00622-f007:**
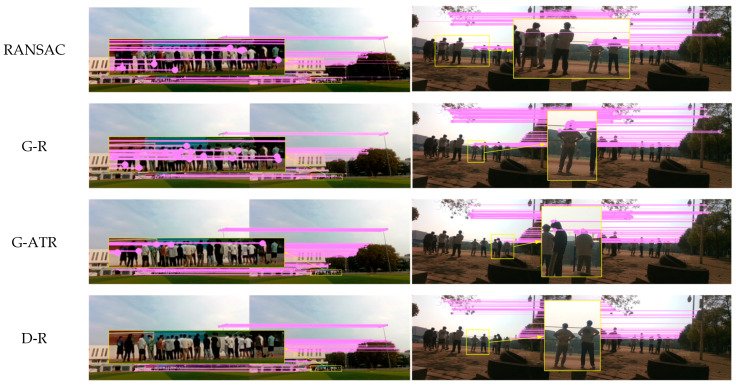
Comparison of feature matching performance between RANSAC, GMS-RANSAC (G-R), GMS-ATRANSAC (G-ATR) and DSSAC-RANSAC (D-R) in self-collected outdoor scenes.

**Figure 8 sensors-25-00622-f008:**
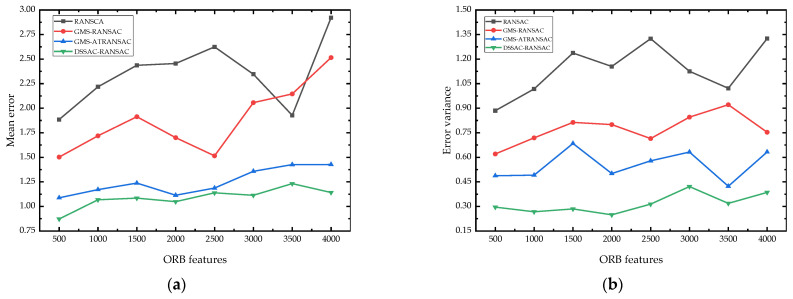
Comparison of the matching mean error (**a**) and error variance (**b**) of RANSAC, GMS-RANSAC, GMS-ATRANSAC and DSSAC-RANSAC at different numbers of feature points.

**Figure 9 sensors-25-00622-f009:**
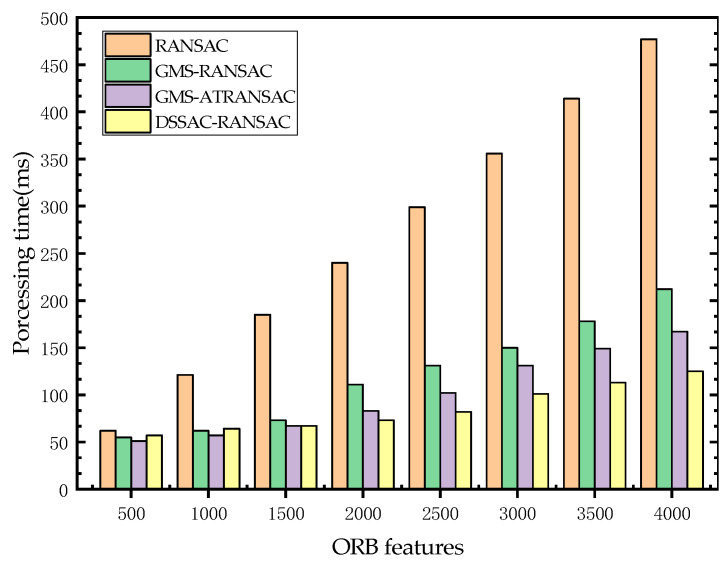
Runtime of four algorithms with different numbers of feature points.

**Figure 10 sensors-25-00622-f010:**
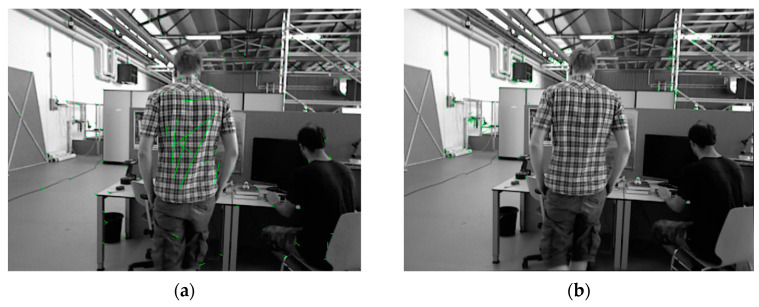
ORB-SLAM2 default initialization method (**a**) and proposed initialization method (**b**) in the TUM indoor dynamic dataset.

**Figure 11 sensors-25-00622-f011:**
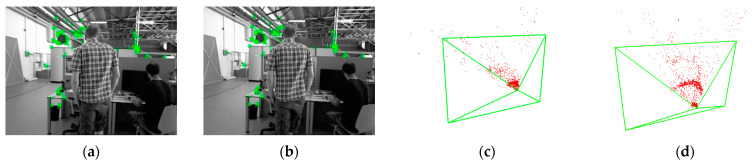
DSSAC-RANSAC applied to feature tracking between adjacent keyframes in ORB-SLAM3, as shown in (**a**,**b**). (**c**) Depth point cloud map of the indoor scene constructed using the default method of ORB-SLAM3. (**d**) Depth point cloud map of the indoor scene constructed by integrating the method proposed in this paper.

**Table 1 sensors-25-00622-t001:** Accuracy and speed comparison of RANSAC (RAN), GMS-RANSAC (G-R), GMS-ATRANSAC (G-ATR) and the proposed algorithm (D-R) when processing 1500 ORB feature points.

Sequences	Metrics	RAN	G-R	G-ATR	D-R
TUM (Indoor dynamic)	$e_{mean}$	2.437	1.913	1.237	1.084
	$e_{Var}$	1.237	0.813	0.684	0.284
	time (ms)	185	73	67	67
KITTI (Outdoor dynamic)	$e_{mean}$	2.372	1.897	1.351	0.957
	$e_{Var}$	1.113	0.901	0.705	0.253
	time (ms)	187	82	70	66
Self-collected (Indoor dynamic)	$e_{mean}$	2.675	2.125	1.411	1.021
	$e_{Var}$	1.352	1.021	0.825	0.275
	time (ms)	178	78	68	58
Self-collected (Outdoor dynamic)	$e_{mean}$	2.771	2.095	1.382	1.102
	$e_{Var}$	1.531	1.103	0.833	0.301
	time (ms)	189	79	63	54

**Table 2 sensors-25-00622-t002:** Accuracy and speed comparison of RANSAC (RAN), GMS-RANSAC (G-R), GMS-ATRANSAC (G-ATR) and the proposed algorithm (D-R) when processing 2000 ORB feature points.

Sequences	Metrics	RAN	G-R	G-ATR	D-R
TUM (Indoor dynamic)	$e_{mean}$	2.455	1.701	1.114	1.048
	$e_{Var}$	1.155	0.801	0.501	0.248
	time (ms)	240	111	83	73
KITTI (Outdoor dynamic)	$e_{mean}$	2.351	1.738	1.125	1.106
	$e_{Var}$	1.038	0.786	0.522	0.283
	time (ms)	237	109	83	71
Self-collected (Indoor dynamic)	$e_{mean}$	2.536	1.835	1.221	0.973
	$e_{Var}$	1.257	0.897	0.613	0.375
	time (ms)	221	107	79	69
Self-collected (Outdoor dynamic)	$e_{mean}$	2.702	1.844	1.432	1.124
	$e_{Var}$	1.215	0.911	0.671	0.331
	time (ms)	217	109	80	71

**Table 3 sensors-25-00622-t003:** Accuracy and speed comparison of RANSAC (RAN), GMS-RANSAC (G-R), GMS-ATRANSAC (G-ATR) and the proposed algorithm (D-R) when processing 2500 ORB feature points.

Sequences	Metrics	RAN	G-R	G-ATR	D-R
TUM (Indoor dynamic)	$e_{mean}$	2.625	1.515	1.187	1.138
	$e_{Var}$	1.325	0.715	0.578	0.314
	time (ms)	299	131	102	82
KITTI (Outdoor dynamic)	$e_{mean}$	2.513	1.419	1.232	1.056
	$e_{Var}$	1.287	0.749	0.613	0.289
	time (ms)	287	134	102	81
Self-collected (Indoor dynamic)	$e_{mean}$	2.633	1.533	1.319	1.121
	$e_{Var}$	1.397	0.813	0.647	0.348
	time (ms)	275	129	100	83
Self-collected (Outdoor dynamic)	$e_{mean}$	2.671	1.601	1.281	1.005
	$e_{Var}$	1.433	0.781	0.634	0.482
	time (ms)	273	129	99	77

**Table 4 sensors-25-00622-t004:** Percentage improvement of DSSAC-RANSAC over RANSAC, GMS-RANSAC and GMS-ATRANSAC in error variance, mean error, and runtime when processing 1500 ORB feature points.

Sequences	$Imprv(e_{mean})$ For RAN (%)	$Imprv(e_{mean})$ For G-R (%)	$Imprv(e_{mean})$ For G-ATR (%)	$Imprv(e_{Var})$ For RAN (%)	$Imprv(e_{Var})$ For G-R (%)	$Imprv(e_{Var})$ For G-ATR (%)	Imprv(time) For RAN (%)	Imprv(time) For G-R (%)	Imprv(time) For G-ATR (%)
TUM (Indoor dynamic)	55.5%	77.0%	12.4%	43.3%	65.0%	58.5%	63.8%	8.2%	0%
KITTI (Outdoor dynamic)	59.7%	77.2%	29.2%	49.6%	77.3%	64.1%	64.7%	19.5%	5.7%
Self- collected (Indoor dynamic)	61.8%	79.7%	27.6%	52.0%	73.1%	66.7%	71.4%	25.6%	14.7%
Self- collected (Outdoor dynamic)	60.2%	80.3%	20.2%	47.4%	72.7%	63.7%	71.4%	31.6%	14.3%
AVG.	59.3%	78.6%	22.4%	48.1%	72.0%	63.3%	67.8%	21.2%	8.7%

**Table 5 sensors-25-00622-t005:** Percentage improvement of DSSAC-RANSAC over RANSAC, GMS-RANSAC and GMS-ATRANSAC in error variance, mean error, and runtime when processing 2000 ORB feature points.

Sequences	$Imprv(e_{mean})$ For RAN (%)	$Imprv(e_{mean})$ For G-R (%)	$Imprv(e_{mean})$ For G-ATR (%)	$Imprv(e_{Var})$ For RAN (%)	$Imprv(e_{Var})$ For G-R (%)	$Imprv(e_{Var})$ For G-ATR (%)	Imprv(time) For RAN (%)	Imprv(time) For G-R (%)	Imprv(time) For G-ATR (%)
TUM (Indoordynamic)	57.3%	38.4%	6%	78.5%	69.0%	50.0%	69.6%	34.2%	12.0%
KITTI (Outdoor dynamic)	53.0%	36.4%	2%	72.7%	64.0%	45.8%	70.0%	34.9%	14.5%
Self- collected (Indoor dynamic)	61.6%	47.0%	20.3%	70.2%	58.2%	38.8%	68.8%	35.5%	12.3%
Self- collected (Outdoor dynamic)	58.4%	39.0%	21.5%	72.8%	63.7%	50.7%	67.3%	34.9%	11.2%
AVG.	57.6%	40.2%	12.5%	73.6%	63.7%	46.3%	68.9%	34.9%	12.5%

**Table 6 sensors-25-00622-t006:** Percentage improvement of DSSAC-RANSAC over RANSAC, GMS-RANSAC and GMS-ATRANSAC in error variance, mean error, and runtime when processing 2500 ORB feature points.

Sequences	$Imprv(e_{mean})$ For RAN (%)	$Imprv(e_{mean})$ For G-R (%)	$Imprv(e_{mean})$ For G-ATR (%)	$Imprv(e_{Var})$ For RAN (%)	$Imprv(e_{Var})$ For G-R (%)	$Imprv(e_{Var})$ For G-ATR (%)	Imprv(time) For RAN (%)	Imprv(time) For G-R (%)	Imprv(time) For G-ATR (%)
TUM (Indoor dynamic)	56.6%	24.9%	4.1%	76.3%	56.1%	45.7%	72.6%	37.4%	19.6%
KITTI (Outdoor dynamic)	58.0%	25.6%	14.3%	77.5%	61.4%	52.9%	71.8%	39.6%	20.6%
Self- collected (Indoor dynamic)	57.4%	26.9%	15.0%	75.1%	57.2%	46.2%	69.8%	35.7%	17%
Self- collected (Outdoor dynamic)	62.4%	37.2%	21.5%	66.4%	38.3%	24.0%	71.8%	40.3%	22.2%
AVG.	58.6%	28.7%	13.7%	73.8	53.3%	42.2%	71.5%	38.3	19.9%

**Table 7 sensors-25-00622-t007:** The average percentage improvement in error variance, mean error, and runtime for DSSAC-RANSAC compared to RANSAC, GMS-RANSAC, and GMS-ATRANSAC when processing 1500, 2000, and 2500 ORB feature points.

ORB Feature Points	$Imprv(e_{mean})$ For RAN (%)	$Imprv(e_{mean})$ For G-R (%)	$Imprv(e_{mean})$ For G-ATR (%)	$Imprv(e_{Var})$ For RAN (%)	$Imprv(e_{Var})$ For G-R (%)	$Imprv(e_{Var})$ For G-ATR (%)	Imprv(time) For RAN (%)	Imprv(time) For G-R (%)	Imprv(time) For G-ATR (%)
1500	59.3%	78.6%	22.4%	48.1%	72.0%	63.3%	67.8%	21.2%	8.7%
2000	57.6%	40.2%	12.5%	73.6%	63.7%	46.3%	68.9%	34.9%	12.5%
2500	58.6%	28.7%	13.7%	73.8	53.3%	42.2%	71.5%	38.3%	19.9%
AVG.	58.5%	49.2%	16.2%	65.2%	63.0%	50.6%	69.4%	31.5%	13.7%

## Data Availability

Data are contained within the article.

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
