# Peer review of "Research on a Density-Based Clustering Method for Eliminating Inter-Frame Feature Mismatches in Visual SLAM Under Dynamic Scenes"

_sensors, 2025, doi:10.3390/s25030622_

Round 1
Reviewer 1 Report
Comments and Suggestions for Authors
1. There are many typos in the text and graphs.
2. It is difficult to visually discern how much the matching points have decreased in Figures 1 and 3. Supplementation is needed to enable more intuitive comparison.
3. Figures 7 and 8 show the matching point comparison results for four algorithms: RANSAC, GMS-RANSAC, GMS-ATRANSAC, and DSSAC-RANSAC, while in Figures 4 to 6, there are only three: RANSAC, GMS-RANSAC, and DSSAC-RANSAC. I think it would be a good idea to add matching point pictures from GMS-ATRANSAC as well.
4. For Figures 4 to 6, it would be good if the comparison results of matching or mismatching points could be displayed more intuitively.
5. It would be good if GMS-ATRANSAC results were added to Tables 1 and 2.
6. In the results of Tables 1 and 2, the performance comparison was made only for 1500 ORB features. I wonder if there is a special reason for the performance comparison for only 1500 ORB features. If there is no special reason, it would be better if performance evaluation was performed on ORB features in the range of 500 to 4000, as shown in Figures 7 and 8.
7. The title of Section 4.3.2 is incorrect.
8. It would be good to express the points in Figure 9 more clearly.
9. If (c) in Figure 10 is a point cloud map for (a) and (b) spaces when applying the DSSAC-RANSAC algorithm, it is generated when existing algorithms (RANSAC, GMS-RANSAC, GMS-ATRANSAC) that cannot remove mismatching points are used. It would be a good idea to add the results so that we can compare point cloud maps. This is believed to be the most important content to prove the validity of the method of improving the tracking performance of the SLAM algorithm through efficient removal of mismatching points proposed in this paper.
Reviewer 2 Report
Comments and Suggestions for Authors
In this paper, the authors propose a novel image feature matching algorithm named DSSAC-RANSAC, which aims to address the issue of feature point mismatches in visual SLAM systems under dynamic scenes.By combining the strengths of DBSCAN and RANSAC, the algorithm employs density-based clustering and geometric model fitting to effectively eliminate feature mismatches.Experiments conducted on indoor and outdoor dynamic datasets demonstrate that the proposed methods with high accuracy and efficiency in eliminating feature mismatches.The practical value in visual SLAM makes this paper potentially acceptable with minor modifications. Here are some suggestions:
1. The paper lacks an overall scheme framework diagram, which is recommended to be supplemented.
2. It is recommended to elaborate on the clustering process of the DSSAC algorithm, including how to determine clustering parameters and how to handle boundary points and noise points.
3. How to coordinate the parameter settings of the DSSAC and RANSAC algorithms.
4. The subtitle of section 4.3.2 is incorrect and needs to be revised.
Comments on the Quality of English Language
None
Round 2
Reviewer 1 Report
Comments and Suggestions for Authors
It is believed that appropriate revisions and supplements have been made to the paper.